# Vertebrate-*Aedes aegypti* and *Culex quinquefasciatus* (Diptera)-arbovirus transmission networks: Non-human feeding revealed by meta-barcoding and next-generation sequencing

José Guillermo Estrada-Franco[1]☯, Nadia A. Fernández-Santos[1]☯, Adeniran A. Adebiyi[1]☯, María de J. López-López[1], Jesús A. Aguilar-Durán[1], Luis M. Hernández-Triana[2], Sean W. J. Prosser[3], Paul D. N. Hebert[3], Anthony R. Fooks[2], Gabriel L. Hamer[4], Ling Xue[5], Mario A. Rodríguez-Pérez[1]*

**1** Instituto Politécnico Nacional, Centro de Biotecnología Genómica, Ciudad Reynosa, Tamaulipas, México, **2** Animal and Plant Health Agency, Woodham Lane, Addlestone, Surrey, United Kingdom, **3** Centre for Biodiversity Genomics, University of Guelph, Ontario, Canada, **4** Department of Entomology, Texas A&M University, College Station, Texas, United States of America, **5** College of Mathematical Sciences, Harbin Engineering University, Harbin, Heilongjiang, P.R. China

☯ These authors contributed equally to this work.

\* drmarodriguez@hotmail.com, mrodriguez@ipn.mx

## Abstract

### Background

*Aedes aegypti* mosquito-borne viruses including Zika (ZIKV), dengue (DENV), yellow fever (YFV), and chikungunya (CHIKV) have emerged and re-emerged globally, resulting in an elevated burden of human disease. *Aedes aegypti* is found worldwide in tropical, sub-tropical, and temperate areas. The characterization of mosquito blood meals is essential to understand the transmission dynamics of mosquito-vectored pathogens.

### Methodology/principal findings

Here, we report *Ae. aegypti* and *Culex quinquefasciatus* host feeding patterns and arbovirus transmission in Northern Mexico using a metabarcoding-like approach with next-generation deep sequencing technology. A total of 145 *Ae. aegypti* yielded a blood meal analysis result with 107 (73.8%) for a single vertebrate species and 38 (26.2%) for two or more. Among the single host blood meals for *Ae. aegypti*, 28.0% were from humans, 54.2% from dogs, 16.8% from cats, and 1.0% from tortoises. Among those with more than one species present, 65.9% were from humans and dogs. For *Cx. quinquefasciatus*, 388 individuals yielded information with 326 (84%) being from a single host and 63 (16.2%) being from two or more hosts. Of the single species blood meals, 77.9% were from dogs, 6.1% from chickens, 3.1% from house sparrows, 2.4% from humans, while the remaining 10.5% derived from other 12 host species. Among those which had fed on more than one species, 11% were from dogs and humans, and 89% of other host species combinations. Forage ratio analysis revealed

**Data Availability Statement:** All raw data has been submitted to the SRA (http://www.ncbi.nlm.nih.gov/bioproject/600226) under accession PRJNA600226.

**Funding:** This work was performed under the auspices of CONACyT-MEXBOL (No. 295569) to MARP; IPN (SIP 20181120 and 20195706) to MARP, and a TAMU-CONACYT Collaborative Research Grant Program (2018-041-1) to MARP and GLH. AAA holds a doctoral scholarship from Consejo Nacional de Ciencia Y Tecnología (CONACYT), Mexico (291137/457158). Funding for LMHT and ARF was provided by the Department for Environment Food and Rural Affairs (DEFRA), Scottish Government and Welsh Government through grants SV3045 and the EU Framework Horizon 2020 Innovation Grant, European Virus Archive (EVAg, grant no. 653316). Funding for JGEF was provided by IPN (SIP 20196759, 20200873, and 20202442). PNDH would like to thank Canada First Research Excellence Fund and the government of Canada through Genome Canada and the Ontario Genomics Institute to the International Barcode of Life Project which also aided the work. The funders had no role in study design, data collection and analysis, decision to publish, or preparation of the manuscript.

**Competing interests:** The authors have declared that no competing interests exist.

dog as the most over-utilized host by *Ae. aegypti* (= 4.3) and *Cx. quinquefasciatus* (= 5.6) and the human blood index at 39% and 4%, respectively. A total of 2,941 host-seeking female *Ae. aegypti* and 3,536 *Cx. quinquefasciatus* mosquitoes were collected in the surveyed area. Of these, 118 *Ae. aegypti* pools and 37 *Cx. quinquefasciatus* pools were screened for seven arboviruses (ZIKV, DENV 1–4, CHIKV, and West Nile virus (WNV)) using qRT-PCR and none were positive (point prevalence = 0%). The 95%-exact upper limit confidence interval was 0.07% and 0.17% for *Ae. aegypti* and *Cx. quinquefasciatus*, respectively

## Conclusions/significance

The low human blood feeding rate in *Ae. aegypti*, high rate of feeding on mammals by *Cx. quinquefasciatus*, and the potential risk to transmission dynamics of arboviruses in highly urbanized areas of Northern Mexico is discussed.

## Author summary

Elucidating arbovirus-vector-host contact networks is critical to understand and control mosquito-borne virus transmission, including pathogens such as ZIKV, DENV 1–4, CHIKV, and WNV. Here, we report the results of metabarcoding of blood meals of two primary pathogen mosquito vectors, *Aedes aegypti* and *Culex quinquefasciatus*. We found limited human blood feeding by *Ae. aegypti* and high preference for feeding on mammals by *Cx. quinquefasciatus*. Interestingly, blood meal analysis revealed dogs as the most utilized host for both vector species suggesting the potential for zooprophylaxis for human-amplified urban arboviruses. Pools of these vector species were tested for seven arboviruses and all were negative. We calculated vectorial capacity to discuss the potential risk and transmission dynamics of pathogens transmitted by these two important vectors in an urban location in Northern Mexico.

## Introduction

Mosquitoes (Culicidae) are a diverse taxonomic group found worldwide except Antarctica [1]. Most mosquito species require a blood meal for egg maturation, and this behaviour plays a central role in the transmission of pathogens that cause millions of deaths annually. The most common arboviral vector, *Aedes aegypti* (L.) often takes more than one blood meal during a single egg maturation cycle [2], while *Culex quinquefasciatus* also transmits numerous viruses through its feeding habit. These feeding requirements elicit anthropophilic and zoophilic behaviours that result in complex blood-feeding patterns [3]. Understanding these interactions by characterizing mosquito blood meals is an important tool in unravelling host-vector relationships and associated diseases transmission dynamics [4]. Furthermore, analysing the vector host range by estimating the frequency at which vector populations feed on particular hosts can be used as a proxy for assessment of human exposure and in estimation of vectorial capacity [5].

Various techniques have been employed for the analysis of mosquito blood meals such as ELISA or precipitin tests [6,7], and PCR-based approaches [8]. Apart from the PCR-Sanger sequencing, such methods are often limited to identifying only a certain protein or region of a

gene from a specific host, rendering them amenable to only absence/presence detection of *a priori* identified blood meal hosts [3]. Even with PCR-based methods, results can be limited due to a lack of primers that amplify a wide range of vertebrate hosts [9]. Furthermore, when PCR amplification is successful, mixed-feeding events can be missed by direct Sanger sequencing [10]. High-throughput next generation sequencing technology can overcome this barrier, but to our knowledge this application has only been applied to *Anopheles* spp. [3].

The mitochondrial cytochrome *c* oxidase subunit I (COI) gene has been widely adopted for species identification because it allows species identification through the assignment of a Barcode Index Number [11,12]. It has the additional advantage that reference sequences are available for more than 700,000 animal species, far broader coverage than for other mitochondrial genes, such as *cyt b* [9] and 16S rRNA. DNA metabarcoding is a rapidly evolving method for biodiversity assessment of bulk samples using the COI gene and next-generation sequencing. It has been used in diverse applications including biodiversity monitoring, reconstruction of paleo-communities, and animal gut contents [13]. Given the importance of correctly identifying the preferred vector when examining host-vector contact [a concept expressed by the Human Blood Index (HBI)], which is defined as the proportion of insects in a vector population with human blood [14], we investigated the host feeding pattern of *Ae. aegypti* and *Cx. quinquefasciatus* using a DNA metabarcoding approach at sites near Reynosa, Mexico, a USA-Mexico border city. In addition, we tested for the presence of key arboviruses which are known to circulate in the region.

## Methodology

### Ethics statement

The present study was approved by the Bioethical Committee of the Faculty of Medicine at the Universidad Autónoma del Estado de México (Approval No. 011/2015) to JGEF.

### Study sites and sample collection

Mosquitoes were collected at sites in Pedro J. Méndez neighbourhood (26˚ 1' 3.576" N, 98˚ 16' 30.18" W), Reynosa, Tamaulipas, in Northern Mexico (Fig 1). The neighborhood is semi-urban characterized by poor infrastructure. Intrastreet roads are unpaved, and houses are mostly uncompleted or unplastered. Waste management practices are poor and streets are often flooded after rainfall. The population of the area is about 7,031 people over a *ca*. 6,016 square/meters comprising an estimated 1,750 households.

Sampling was carried out in the rainy (May to July 2018) and dry (September to November 2018) seasons in 16 households, although some received limited collections due to difficulties in access. Four traps were used for collection in each household. Host-seeking mosquitoes were collected using Biogents Sentinel 1 traps (Biogents AG, Germany); one trap was placed outside each home on a weekly basis for a 24-hour period. Gravid mosquitoes were collected using the Autocidal Gravid Ovitraps (AGO) traps (Springstar) that were checked weekly [15]. Indoor resting and host-seeking populations were collected using a CDC backpack aspirator (John W. Hock Company, No. 1412.01, Gainesville, FL) for about 3-min; outdoor resting and blood-engorged mosquitoes were collected using the back-pack aspirator that collected all resting mosquitoes inside a 1 m$^3$ wooden box opened on one side with both the outside and the inside painted red. Collected mosquitoes were preserved on ice in the field and transported to the laboratory, where they were stored at -80˚C until processing. Sample identification and sorting were carried out on a BioQuip chill plate (BioQuip, USA). Mosquito morphological identification was conducted using standard keys [1,16]. Female mosquitoes were further classified into different feeding stages and only engorged mosquitoes of Sella scale 2 (freshly fed)

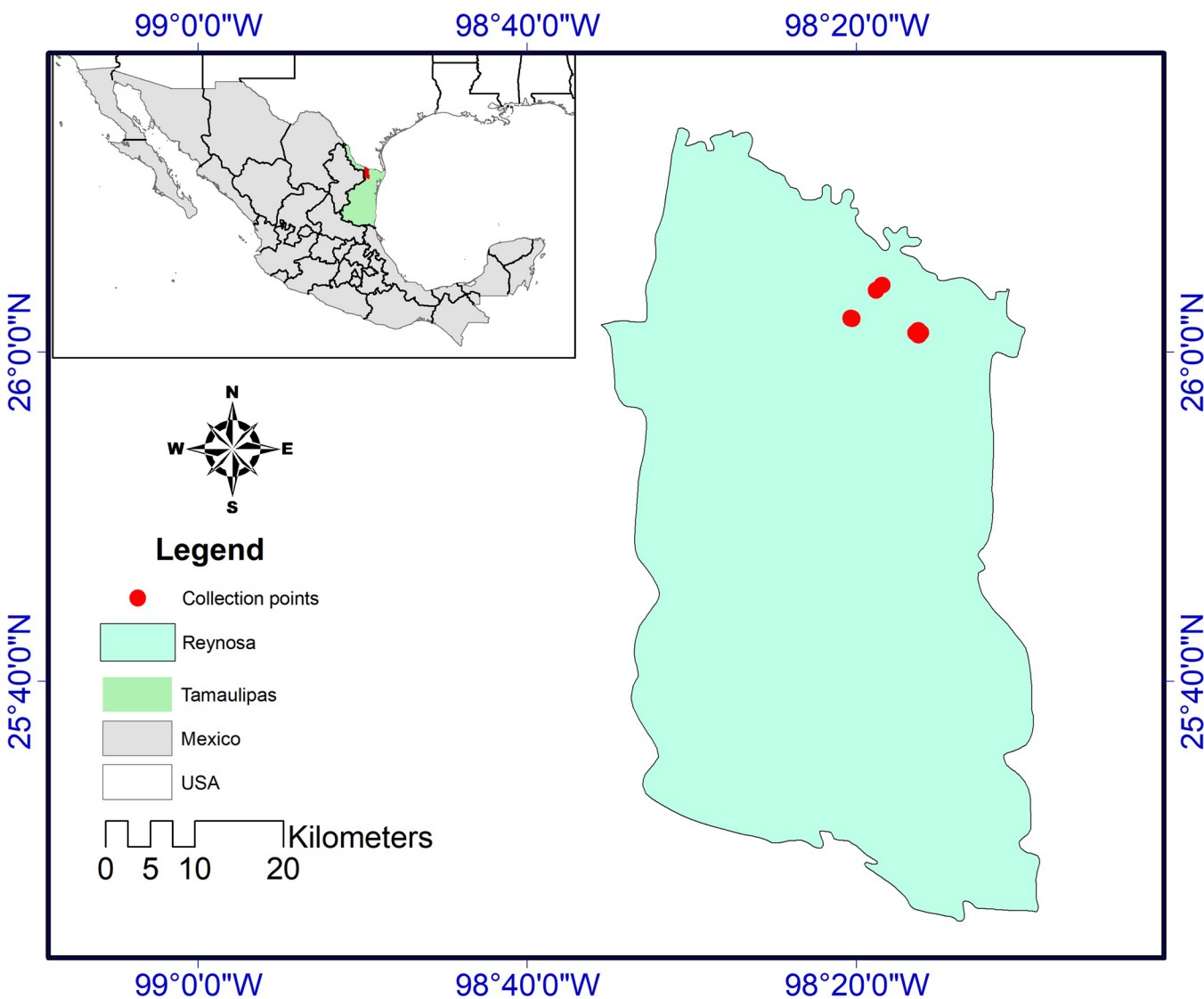

**Fig 1. Study site and location of traps in Pedro J. Mendez neighbourhood in Reynosa, Northern Mexico.** Fig 1 is an original map, created by co-author AAA and is available for use under the CC-BY license. ESRI 2011. ArcGIS Desktop: Release 10. Redlands, CA: Environmental Systems Research Institute.

and 3 (late stage fed) were processed for the blood meal analysis [17]. Engorged mosquitoes were placed individually into nuclease-free 1.5 mL micro-centrifuge tubes, labelled with an identification number and stored at -80$^0$ C for further processing.

The study area was residential and the objective of the study is to understand vector behavior in this residential area. The household selection was random and largely dependent on permission granted by house owners as well as availability to allow entrance on their properties to check traps and retrieved mosquitos. Mosquito collection was done outdoors in the yard of these properties using AGO traps, Biogents Sentinel 1 traps (BGS1), and the resting boxes. Only the backpack aspiration was done indoors. Also, the nature of the study area, where all of the houses shared fences, provided opportunity to sample mosquitoes within a 50-meter radius of each outdoor trap, as this is within the flight range of mosquitoes. Furthermore, with a sampling duration of 5 months, we have a strong reason to believe there is enough representation of the mosquito population in this area. We did not attempt to quantify the behavior of the

household occupants and how much time was spent inside versus outside. However, anecdotally, we observed a significant proportion of their activities, including cooking, grilling, and evening family gatherings, were done outside. Permeability of houses to mosquitoes as well was not measured. Given mosquitoes can have a flight range of up to 100 m post-feeding [18] and species of *Aedes* and *Culex* have 2–5 km flight range [19], we do not think the spatial distribution of the households might have significantly impacted the bloodmeal feeding pattern observed.

## DNA extraction and purification

Blood-fed *Ae. aegypti* and *Cx. quinquefasciatus* were sent to the Canadian Centre for DNA Barcoding, Centre for Biodiversity Genomics, University of Guelph (Guelph, ON, Canada) for DNA extraction and sequencing. The frozen mosquitoes were incubated in liquid nitrogen for one minute to ensure that the blood meal did not thaw and leak during removal of the abdomen. The abdomens were removed with forceps and placed directly into 60 μL of lysis buffer [700 mM guanidine thiocyanate (Sigma), 30 mM EDTA pH 8.0 (Fisher Scientific), 30 mM Tris-HCl pH 8.0 (Sigma), 0.5% Triton X-100 (Sigma), 5% Tween-20 (Fluka Analytical), and 2 mg/mL of Proteinase K (Promega)], which was then incubated for 18 h at 56°C with gentle shaking. DNA purification employed the glass fiber technique described by Ivanova *et al.*[20]. The lysate was mixed with two volumes (120 μL) of binding mix [3 M guanidine thiocyanate, 10 mM EDTA pH 8.0, 5 mM Tris-HCl pH 6.4, 2% TritonX-100, and 50% ethanol] and applied to a well of a 96-well silica membrane plate (PALL), 700 μL at a time. The plate was centrifuged at 5000g for 5 min after each application. The membrane was washed once with 180 μL of protein wash buffer [1.56 M guanidine thiocyanate, 5.2 mM EDTA pH 8.0, 2.6 mM Tris-HCl pH 6.4, 1.04% Triton X-100, and 70% ethanol] and centrifuged at 5000g for 2 min. The plate was then washed a second time with 750 μL of wash buffer [50 mM NaCl (Fisher Scientific), 0.5 mM EDTA pH 8.0, 10 mM Tris-HCl pH 7.4, and 60% ethanol] and centrifuged at 5000g for 5 min. Centrifugation was repeated a second time to remove residual wash buffer. All traces of wash buffer were removed by incubating the plate at 56°C for 30 min. To release DNA from the silica membrane, 40 μL of elution buffer [10 mM Tris-HCl, pH 8.0, pre-warmed to 56°C] was applied directly to the membrane and allowed to incubate at room temperature for 1 min. DNA was eluted from the membrane (into a clean plate) via centrifugation at 5000g for 5 min. A negative control (lysis buffer only) was processed in parallel with the samples.

## PCR amplification and sequencing

Host DNA was prepared for deep sequencing using a two-stage fusion primer approach [13] with vertebrate-specific primers. Two microliters of DNA extract was added to a PCR reaction consisting of 6.25 μL of 10% D-(+)-trehalose dihydrate (Fluka Analytical), 2.0 μL of Hyclone ultra-pure water (Thermo Scientific), 1.25 μL of 10X Platinum Taq buffer (Invitrogen), 0.625 μL of 50 mM MgCl2 (Invitrogen), 0.125 μL of each 10 μM primer, 0.0625 μL of 10 mM dNTP (KAPA Biosystems), and 0.060 μL of 5U/lL Platinum Taq DNA Polymerase (Invitrogen) for a total reaction volume of 12.5 μL. The primers were a cocktail of BloodmealF1_t1, BloodmealF2_t1, VR1_t1, VR1d_t1, and VR1i_t1 (S1 Table), which were tailed with M13F and M13R sequences and targeted a 185 bp fragment of the COI barcode region. The BloodmealF primers were designed in this study to amplify DNA from common vertebrates used by mosquitoes, not to amplify insect DNA including mosquito, and to amplify short fragments (<200 bp) given the nature of degraded DNA in blood meals. Thermocycling consisted of an initial denaturation at 95°C for 2 minutes, 60 cycles of 95°C for 40 seconds, 56°C for 40 seconds, and 72°C for 30 seconds, and a final extension at 72°C for 5 minutes. PCR products

were visualized on a 2% E-gel (Invitrogen) to confirm amplification, and then diluted two-fold with sterile water.

The diluted products were then used as a template for a second round of PCR using M13F primers tailed with Ion Xpress universal molecular identifier (UMI) tags and the Ion Torrent "A" sequencing adapter, and M13R primers tailed with the Ion Torrent trP1 sequencing adapter (S1 Table). Reaction components for the second round of PCR were identical to the first and thermocycling consisted of an initial denaturation at 95˚C for 2 minutes, 5 cycles of 95˚C for 40 seconds, 45˚C for 40 seconds, and 72˚C for 30 seconds, 35 cycles of 95˚C for 40 seconds, 51˚C for 40 seconds, and 72˚C for 30 seconds, and a final extension at 72˚C for 5 minutes. The products from the second round of PCR were pooled in equal volumes and purified by mixing 400 μL of pooled product with 400 μL of purification beads (Aline Biosciences, Woburn, MA, USA) for a ratio (v:v) of 1:1 (beads:DNA). The mixture was incubated at room temperature for eight minutes to allow the DNA to bind to the beads, after which the beads were pelleted on a magnet. The supernatant was discarded, and the pellet was washed three times with 1 mL of freshly prepared 80% ethanol and then air dried for 10 minutes. The purified product was eluted from the beads by resuspending the dried pellet in 200 μL of sterile water, pelleting the beads, then carefully transferring 180 μL of the supernatant to a clean 1.5 mL tube. The purified product was quantified using a Qubit 2.0 fluorometer and adjusted to 26 pM with sterile water. The 26 pM library was prepared for sequencing and loaded onto a 530-chip using an Ion Chef automated platform (Thermo Scientific) with standard 400 bp chemistry. Deep sequencing was performed on an Ion Torrent S5 using 850 flows.

## Bioinformatics analysis

Following sequencing, raw sequence reads were automatically assigned to samples (via the UMI tags) by the Torrent Browser suite. The reads were filtered based on a minimum quality value of QV20 and a minimum read length of 225 bp using Sickle (github.com/ucdavis-bioinformatics/sickle). Adapter and primer sequences were removed using CutAdapt [21]. Since the forward primer should be readily visible in the reads, any reads in which the forward primer was not recognized were discarded to ensure that only high-quality reads were included in the final dataset. Additionally, trimmed reads were only retained if they were between 150–200 bp. Reads passing these filters were collapsed into unique sequences (http://hannonlab.cshl.edu/fastx_toolkit/index.html) while retaining original read counts. The collapsed sequences were then clustered into operational taxonomic units (OTUs) using 97% identity [22], and OTU sequences were retained only if they were less than 220 bp in length and composed of at least 100 reads. Each OTU sequence was used to query (BLAST) a custom database composed of global vertebrate COI sequences downloaded from BOLD. The resulting BLAST hits were filtered to retain only those with a minimum match of 95% identity and 100 bp of coverage between the queried sequence and a reference sequence. Furthermore, identifications were only considered reliable if supported by at least 50 original reads. All raw data has been submitted to the SRA under accession PRJNA600226.

## Animal surveys

In order to estimate the relative abundance of avian species in the trapping areas, a bird survey was conducted as described by Brugman [17] in eight points approximately 50 meters apart between 08:00 and 10:00 hrs. when the avian population will still be relatively more active. Observations were made at each point for 10 minutes, and other observed vertebrates, including horse and dogs, were also noted. We also conducted individual household censuses of vertebrates in the trapping area using a questionnaire. An adult in each household was asked for

the number of dogs, cats, chickens, and other animals in the household. Where possible, study personnel conducted headcounts of the animals for each household. The combined total number of animals for each species recorded was used to estimate the abundance of common vertebrates in the trapping area [17].

## Viral extraction and detection by qRT-PCR

Non-engorged *Ae. aegypti* mosquitoes were tested for arboviruses including the four serotypes of DENV, ZIKV, and CHIKV. Mosquito pools were processed according to standard procedures [23]. In brief, each mosquito pool was homogenized in 600 μL of Hank's buffer solution (Gibco cat. #14170112) with a single bead and centrifuged for 5 minutes at 18000 g. RNA extraction was carried out using 100 μL of the supernatant with a MagMax CORE Nucleic Acid Purification Kit (ThermoFisher, Waltham, MA) following the manufacturer's instructions. For a subset of samples we quantified nucleic acid recovered using a spectrophotometer (Epoch, BioTek Instruments, Inc.). DENV was tested for using the PathID Multiplex one step (Applied Biosystems) and Qiagen QuantiTect probe reverse transcriptase polymerase chain reaction (RT-PCR) kits with the primers DENV_F, DENV_R1-3, DENV_R4, and DENV_P probe described by Alm et al. [23]. DENV strains used as positive controls were provided by the University of Texas Medical Branch. We added 5 μL of RNA to a 20 μL master mix containing 12.5 μL RT-PCR buffer, 0.1125 μL each of the primers, 0.2 μL of DENV_P probe, 0.25 μL enzyme, and RNase free water. Cycling conditions were as follows: reverse transcription cycle at 50˚C for 30 minutes, followed by an RT-enzyme inactivation cycle at 95˚C for 20 minutes, then 45 cycles of 95˚C for 3 seconds and 60˚C for 30 seconds. CHIKV and ZIKV were tested for using a multiplex RT-qPCR described by Lanciotti *et al.* [24,25]. Briefly, 5 μL of RNA was added to a master mix containing 0.3 μM of each probe, 2 μL of each of the four primers, 2.5 μL of enzyme mix and RNase free water. Zika PRVABC59 RNA lysate and CHIKV R80422a [23] RNA lysate received from the CDC were used as positive controls. Cycling conditions were as follows: reverse transcription cycle at 48˚C for 10 minutes, followed by an enzyme inactivation cycle at 95˚C for 10 minutes, then 40 cycles of 95˚C for 15 seconds, and 60˚C for 1 minute. All samples with Ct values >38 were considered negative [22,23]. *Cx. quinquefasciatus* was tested for WNV using the primers WNENV-forward (TCAGCGATCTC TCCACCAAAG) and WNENV-reverse (GGGTCAGCACGTTTGTCA-TTG) and the probe WNENV-probe as described by Hamer *et al.* [26].

## Data analysis

We estimated the forage ratio (FR), defined as the frequency at which a mosquito selects a vertebrate host over other available vertebrate hosts, by dividing the observed frequency of blood meals by the expected frequency of blood meals of a given species [27,28]. FR = s/a where s = the percentage of female mosquitoes containing blood of a particular host, and a = percentage of the total available host population represented by that particular host (as determined by the animal survey). A FR of 1.0 indicates mosquitoes are feeding on hosts in equal proportion to availability, whereas values >1.0 indicate over-utilization and values <1.0 indicate under-utilization. The human blood index (HBI) was calculated as the sum of human blood meals for a mosquito species divided by the sum of engorged mosquitoes of a particular species with successful host identification [29].

Virus screening results were analysed using the Bayesian algorithm in the Katholi´s poolscreen program as previously described [30]. The 95% exact upper limit confidence interval (95%-ULCI) for point prevalence of arbovirus infection in both vector species populations were estimated.

## Mathematical modeling

Investigating the transmission efficiency and risk of human exposure to *Ae. aegypti*-transmitted viruses are essential steps to developing successful intervention strategies. Thus, vectorial capacity (VC) that incorporates blood feeding behaviour and vector longevity along with vector competence was calculated to determine the intensity of dengue virus transmission with the following equation [31]:

$$VC = \frac{h \cdot b \cdot p^n}{-\ln(p)}$$

where $n$ = extrinsic incubation period; $b$ = vector competence; $p$ = survival rate; and $h$ = blood feeding rate.

We applied the model of Chan and Johansson [32], which relates the extrinsic incubation period (EIP) with temperature as a covariate. The EIP ($n$, in days) was estimated as a function of temperature ($T$) using the following equation: $n(T) = 600(0.3/2\pi)^{0.5} \exp\left(\frac{-0.3(T-5.9)^2}{T}\right)$.

Vector competence (b) was calculated by using the equation of Sallum *et al.* [33]

$$b = \frac{Incidence(t)}{aS_H(t)} \frac{N_H(t)}{N_M(t)} \frac{N_M(t)}{I_M(t)}$$

where Incidence (t) is the number of new dengue cases per unit time, $S_H(t)$ is the number of susceptible humans, $N_M(t)/N_H(t)$ is the ratio of the number of dengue mosquitoes to the number of humans, and $I_M(t)/N_M(t)$ is the prevalence of dengue virus infection in the vector population.

The survival rate, $p$, can be expressed by $e^{-\mu}$ [34]. Besides, according to the experiments by Yang *et al.* [35] in female *Ae. aegypti* over the temperature range of 10.54°C< = T< = 33.41°C, a $4^{th}$ order polynomial function can be used to describe the relationships between the temperature and μ as follows:

$$\mu(T) = 0.8692 - 0.1590T + 0.01116T^2 - 3.408x10^{-4}T^3 + 3.809x10^{-6}T^4.$$

# Results

## Mosquito relative abundance

Mosquito samples were collected a total of 423 times, with each collection event involving checking all four traps during a visit. Empty traps were not recorded. A total of 15,922 individual *Ae. aegypti* and *Cx. quinquefasciatus* mosquitoes were collected (13,761 from May-July and 2,161 from September-November 2018; S2 Table). A total of 881 individual mosquitoes of other species were also collected which summed to 16,733 individuals of all mosquitoes combined and for all trap types. The taxonomic composition included 13 identified species. The most abundant species of mosquitoes were *Ae. aegypti* (n = 5,211; 31.1%) and *Cx. quinquefasciatus* (n = 10,711; 64.0%; S2 Table). The remaining 11 species of mosquitoes were as follows: *Ae. albopictus* (n = 70; 0.41%), *Aedes* spp. (n = 3; 0.02%), *Anopheles pseudopunctipennis* (n = 3; 0.02%), *Cx. salinarius* (n = 3; 0.02%), *Cx. tarsalis* (n = 3; 0.02%), *Cx. nigripalpus* (n = 7; 0.04%), *Cx. interrogator* (n = 24; 0.14%), *Cx. restuans* (n = 59; 0.35%), *Cx. erythrothorax* (n = 6; 0.03%), *Ae. bimaculatus* (n = 1; 0.005%), and *An. quadrimaculatus* (n = 2; 0.01%). The remaining 3.83% (n = 670) of the mosquitoes were unable to be identified morphologically due to lack of identifiable features or damage body parts. Forty percent (n = 6,477) of the *Ae. aegypti* and *Cx. quinquefasciatus* mosquitoes collected were females and 60% (n = 9,445) were males (S2 Table). The mean number of *Ae. aegypti* and *Cx. quinquefasciatus* mosquitoes collected from

the BGS1 traps per collection was 82.83 (± 11.97 SEM; 95%-CIs = 59.1–105.5), significantly higher than any other trap type (Table 1).

## Blood feeding pattern

A total number of 1,044 bloodfed mosquitoes were collected which included 436 *Ae. aegypti* (41.8%) and 608 *Cx. quinquefasciatus* (58.2%). Overall, 170 *Ae. aegypti* (3 collected using backpack aspirators, 52 collected using BGS1 traps, and 115 collected from resting boxes) and 475 *Cx. quinquefasciatus* (2 collected from AGO trap, 10 collected from CDC backpack aspirators, 269 from BGS1 traps, and 194 from resting boxes) were used in the blood meal analysis. Of the 170 *Ae. aegypti* individual mosquitoes examined, 145 (85%) yielded host identifications. Of the 145 successful individuals, 107 (73.8%) exhibited single host feeding and 38 (26.2%) exhibited mixed host feeding (*i.e.* more than one vertebrate species). Of the 107 individuals that fed on a single host, 54.2% (n = 58) fed on dogs (*Canis lupus familiaris*), 28.0% (n = 30) fed on humans (*Homo sapiens*), 16.8% (n = 18) fed on cats (*Felis silvestris silvestris*), and 1.0% (n = 1) fed on red-eared turtle (*Trachemys scripta*) (Fig 2). Of the 38 individuals that fed on mixed hosts, 65.9% (n = 25) fed on dogs and human, 18.4% (n = 7) fed on dogs and cats, 5.3% (n = 2) fed on dogs and cattle, and 2.6% (n = 4) fed on other host combinations (Fig 3 and S3 Table). There was no significant difference (p = 0.9989) in the host identification success between the two Sella scores used: 85.3% with Sella score 2 and 85.2% with Sella score 3.

A lower sequencing success was achieved for *Cx. quinquefasciatus*. Out of the 475 individuals examined, 388 (81.7%) yielded host identifications. Of the 388 individuals, 326 (84.0%) exhibited single host feeding, while 63 (16.2%) exhibited mixed host feeding. Of the 326 individuals that fed on a single host, 77.9% (n = 254) fed on dogs, 6.13% (n = 20) fed on chickens (*Gallus gallus*), 3.1% (n = 10) fed on house sparrows (*Passer domesticus*), 2.4% (n = 8) fed on humans, 2.1% (n = 7) fed on Eurasian collared doves (*Streptopelia decaocto*), 2.1% (n = 7) fed on Virginia opossums (*Didelphis virginiana*), 2.1% (n = 7) fed on mourning dove (*Zenaida macroura*), 0.9% (n = 3) fed on Inca dove (*Columbina inca*), 0.6% (n = 2) fed each on cats and rock dove (*Columba livia*), and the remaining five *Cx. quinquefasciatus* fed on other hosts (0.3% each; Fig 2 and S3 Table). The 63 individuals that fed on mixed hosts fed on a wide range of vertebrate species (n = 20), with the most frequent combinations being dog and human (11.1%, n = 7), dog and house sparrow (9.5%, n = 6), and dog and turkey (*Meleagris gallopavo*; 9.5%, n = 6; Fig 3 and S3 Table). There was no significant difference (p = 0.260) in host identification success between the two Sella score used: 82.5% with Sella score 2 and 77% with Sella score 3.

**Table 1. Average ± SEM of mosquitoes per trap per collection.** The total mosquito counts retrieved from each trap were averaged by the number of collection events that yielded the count, which translates to the number of individuals per trap per collection event.

| | | Resting red box | CDC backpack aspirator | BG Sentinel trap | AGO trap |
|---|---|---|---|---|---|
| | | $\bar{x}$± SEM | $\bar{x}$± SEM | $\bar{x}$± SEM | $\bar{x}$± SEM |
| *Aedes aegypti* | Male | 5.42 ± 1.45 (2.55–8.28) | 0.90 ± 0.49 (-0.10–1.89) | 10.89 ± 1.86 (7.23–14.56) | 0.24 ± 0.05 (0.14–0.34) |
| | Female | 4.25 ± 0.56 (3.13–5.37) | 2.31 ± 0.50 (1.28–3.34) | 14.14 ± 1.80 (10.58–17.69) | 2.80 ± 0.25 (2.31–3.28) |
| *Culex quinquefasciatus* | Male | 15.49 ± 3.87 (7.87–23.14) | 1.17 ± 0.82 (-0.51–2.85) | 36.37 ± 7.15 (22.24–50.50) | 0.65 ± 0.14 (0.37–0.92) |
| | Female | 6.65 ± 1.38 (3.93–9.38) | 1.52 ± 0.43 (0.63–2.41) | 16.18 ± 3.32 (9.62–22.74) | 3.01 ± 0.47 (2.09–3.93) |
| **Overall** | | **32.56 ± 5.86** (20.96–44.16) | **6.07 ± 1.66** (2.67–9.47) | **82.83 ± 11.97** (59.15–105.50) | **6.82 ± 0.60** (5.63–8.01) |

Values in parentheses represent the 95%-confidence intervals.

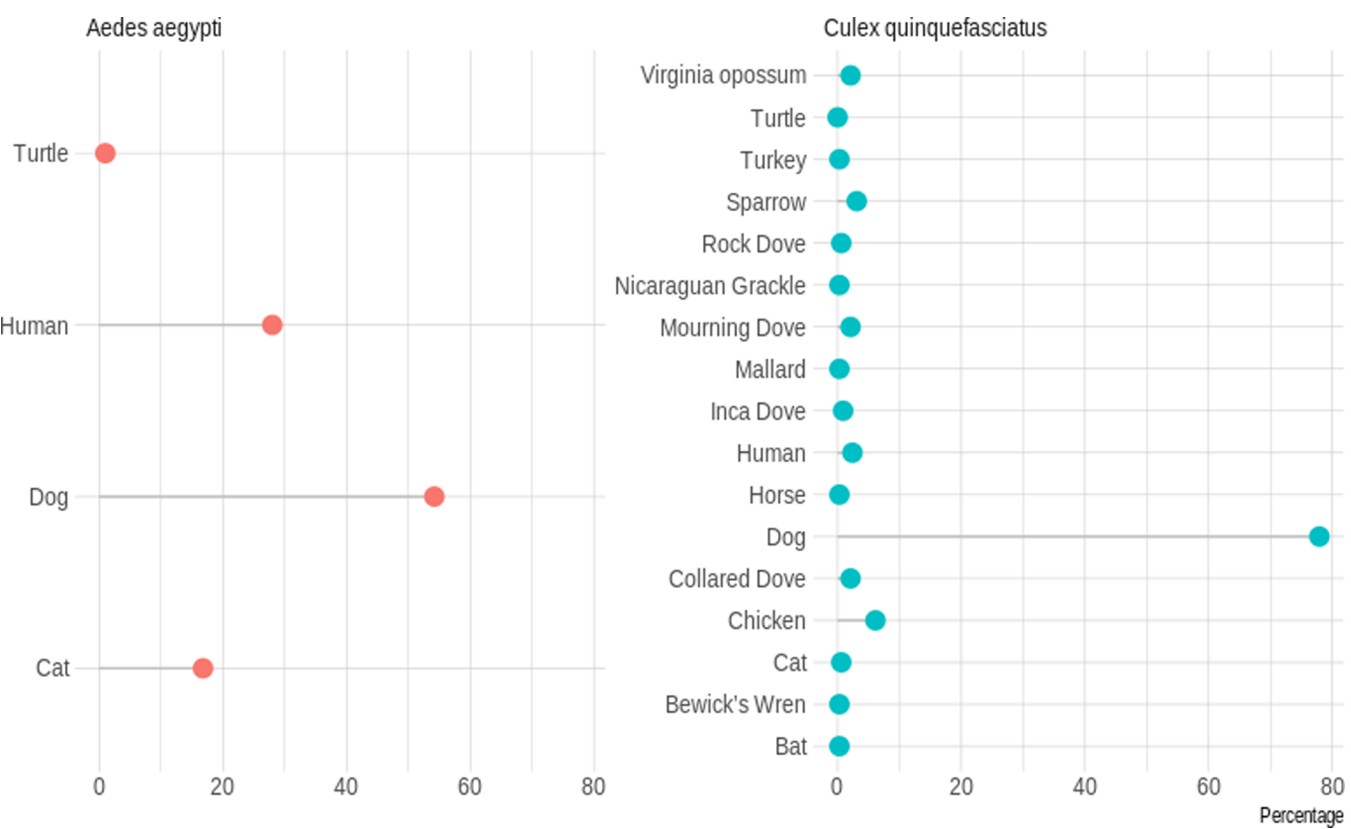

**Fig 2. Host identification for single blood meals from *Aedes aegypti* and *Culex quinquefasciatus*.** *Ae. aegypti* = 25 (14.7%) failed blood meal identification, but 145 individuals were examined; *Cx. quinquefasciatus* = 87 (18.3%) failed identification, but 388 individuals were examined. (Figure percentage is for successful identification).

### Forage ratio and human blood index

Based on our animal survey, each household contained an average of 3.6 (± 0.45 SEM) human inhabitants, and the study area contained an average of 79 humans, 65 dogs, 48 cats, 37 chickens, and 3 pigs (S4 Table). Forage ratios for *Ae. aegypti* and *Cx. quinquefasciatus* were calculated with host availability estimated from these results. The highest forage ratio of *Ae. aegypti* was for dogs (4.34), which was nearly twice as high as the second highest forage ratio for humans (2.16; Table 2). The highest forage ratio of *Cx. quinquefasciatus* was also for dogs (5.64), followed by Virginia opossum (*Didelphis virginiana*; 4.83) and turkey (4.83; Table 3). The human blood index (total number of human blood meals divided by the total number of engorged females with a confirmed result) for *Ae. aegypti* and *Cx. quinquefasciatus* was 39.3% and 4.1%, respectively.

### Mosquito arbovirus RNA testing

A total of 2,941 female *Ae. aegypti* were collected during the study. Out of these, 118 pools were made from 2,545 mosquitoes with an average of 18.4 (± 1.73 SEM) mosquitoes per pool. These were tested and none was found positive (point prevalence = 0) for DENV 1–4, ZIKV and CHIKV using qRT-PCR (positive controls returned positive results, indicating that the assay functioned as expected). Similarly, of the 3,536 female *Cx. quinquefasciatus* specimens, 37 pools, representing 1,057 mosquitoes, was tested for WNV (using qRT-PCR), and all were found to be negative (point prevalence = 0). The associated 95%-ULCI for prevalence of any of

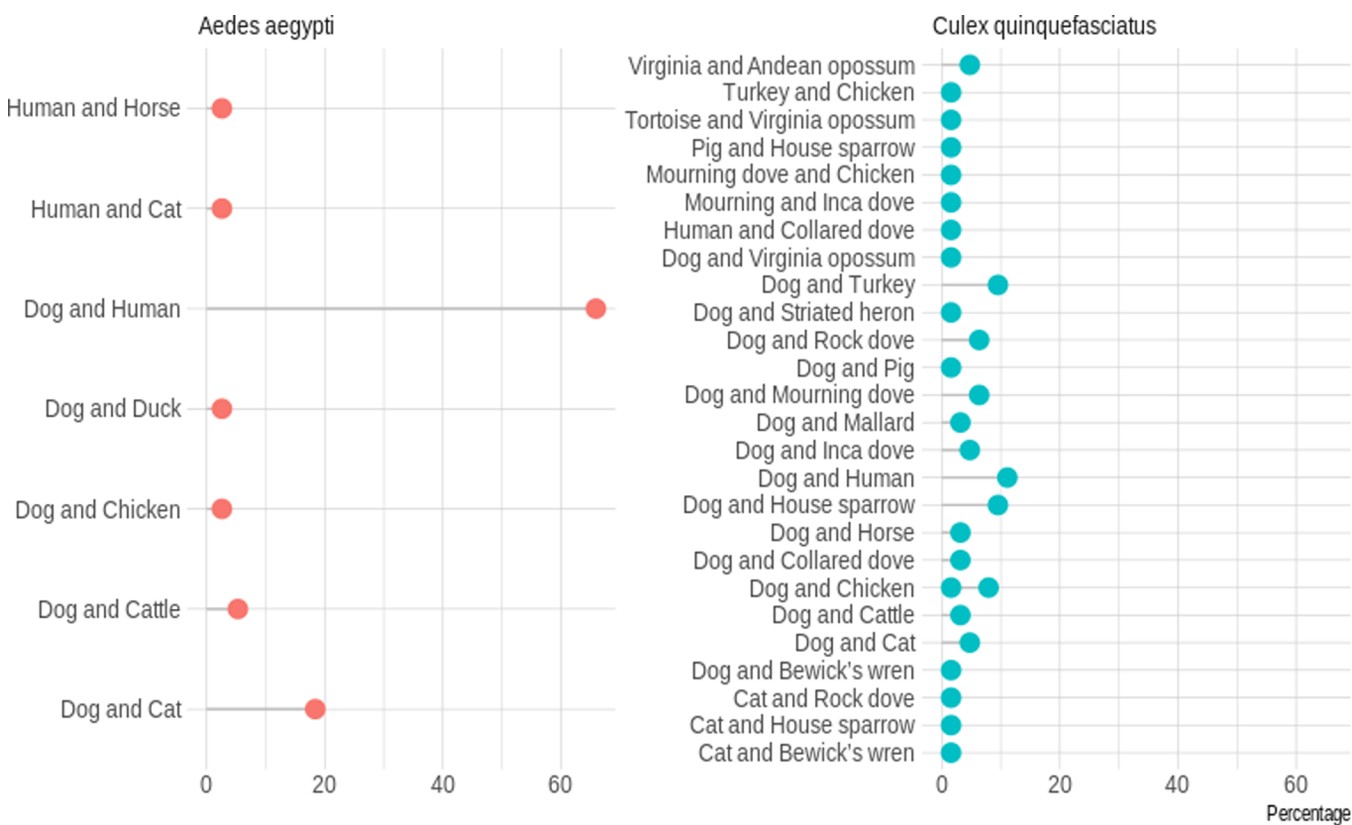

**Fig 3. Host identification for mixed blood meals from *Aedes aegypti* and *Culex quinquefasciatus*.** *Ae. aegypti* = 25 (14.7%) failed blood meal identification, but 145 individuals were examined; *Cx. quinquefasciatus* = 87 (18.3%) failed identification, but 388 individuals were examined. (Figure percentage is for successful identification).

the viruses examined in *Ae. aegypti* was 0.75/1,000 (0.075%) mosquitoes examined, while the associated 95%-ULCI for prevalence of WNV in *Cx. quinquefasciatus* was 1.79/1,000 (0.18%) mosquitoes examined (Table 4).

**Table 2. Forage ratio estimates for *Aedes aegypti* collected in Reynosa, Northern Mexico.**

| Animal host | Number of hosts blood meal identified | % host fed | Estimated host abundance | Forage Ratio | HBI |
|---|---|---|---|---|---|
| Dog | 94 | 51.37% | 11.84% | 4.34 | |
| Human | 57 | 31.15% | 14.39% | 2.16 | 39.31% |
| Cat | 26 | 14.21% | 8.74% | 1.63 | |
| Cow | 2 | 1.09% | - | n/a | |
| Horse | 1 | 0.55% | 1.28% | 0.43 | |
| Muscovy Duck | 1 | 0.55% | 0.18% | 3 | |
| Tortoise | 1 | 0.55% | 0.55% | 1 | |
| Chicken | 1 | 0.55% | 6.74% | 0.08 | |
| **Overall** | **183**[*] | 100.00% | | | |

[*]This is greater than the 145 number of *Ae. aegypti* with successful host identification because each mixed feeding event represented two host species.

n/a = not surveyed.

HBI-Human blood proportion or engorged females containing human blood.

**Table 3. Forage ratio for *Culex quinquefasciatus* collected in Reynosa, Northern Mexico.**

| Vertebrate host | # of host fed | % of host fed | Estimated abundance | Forage ratio |
|---|---|---|---|---|
| Dog | 304 | 66.81% | 11.84% | 5.64 |
| Chicken | 31 | 6.81% | 6.74% | 1.01 |
| House sparrow | 18 | 3.96% | 18.40% | 0.22 |
| Human | 16 | 3.52% | 14.39% | 0.24 |
| Mourning dove | 13 | 2.86% | 0.73% | 3.92 |
| Virginia opossum | 12 | 2.64% | 0.55% | 4.83 |
| Eurasian collared dove | 10 | 2.20% | 4.37% | 0.50 |
| Cat | 8 | 1.76% | 8.74% | 0.20 |
| Rock dove | 8 | 1.76% | 1.09% | 1.61 |
| Turkey | 8 | 1.76% | 0.36% | 4.83 |
| Inca dove | 6 | 1.32% | 15.30% | 0.09 |
| Mallard | 4 | 0.89% | 0.36% | 2.47 |
| Andean opossum | 3 | 0.66% | | |
| Horse | 3 | 0.66% | 1.28% | 0.52 |
| Bewick's wren | 3 | 0.66% | | |
| Cow | 2 | 0.44% | | |
| Pig | 2 | 0.44% | 0.55% | 0.80 |
| Texas tortoise | 1 | 0.22% | 0.55% | 0.40 |
| Northern yellow bat | 1 | 0.22% | | |
| Nicaraguan grackle | 1 | 0.22% | | |
| Striated heron | 1 | 0.22% | | |
| **Overall** | **455**[*] | 100.00% | | |

[*]This is greater than the 388 number of *Cx. quinquefasciatus* with successful host identification because of mixed host feeding, with each mixed feeding representing two host species.

## Mathematical modelling

According to the findings of Rodríguez-Pérez et al. [36], six people were IgM-positive and seroconverted from a total of 77 examined people, therefore the Incidence/$S_H$ = 6/(77−6). The prevalence of dengue virus infection should be varying with time. We assumed that it is a constant due to the absence of such data. Similarly, the ratio of the number of new cases to the number of humans is also assumed to be constant. We derived the ratio of the number of humans to the number of mosquitoes. Namely, 1,527 landing counts in 231 households; then assuming on average four persons per household, yields $N_H/N_M$ = 231*4/1527. The infection rate of DENV in *Ae. aegypti* is 0.47% (3/633) as from Rodríguez-Pérez et al. [36]. The biting rate (*a*) as from Garza-Hernández et al. [37], where a = 1/5, the human biting rate for DENV in Mexico.

**Table 4. qRT-PCR virus screening results of the mosquito pools examined.**

| Mosquito species | Virus screened | Number of pools examined | Pool size range | Number of mosquitoes | qRT-PCR Positive pools | Point prevalence of infected mosquitoes/10,000[*] |
|---|---|---|---|---|---|---|
| *Ae. aegypti* | DENV (1–4), ZIKV, CHIKV | 118 | 1–83 | 2,545 | 0 | 0 (7.51) |
| *Cx. quinquefasciatus* | WNV | 37 | 1–74 | 1,057 | 0 | 0 (17.96) |

[*]The value represents the maximum likelihood point estimate and the value in parentheses represents the upper bound 95%-confidence interval.

Furthermore, the blood feeding rate $h$ be (105/1106) $^*$(57/145) = 3.7% (S2 Table). Here, 105 is the number of fed females and 1,106 is the total caught number of *Ae. aegypti*. According to Figs 2 and 3, 57/145 = 39.3% is the proportion of human blood meals identified. Then 105$^*$57/145 is the number of mosquitoes that will bite human among the total 1,106 mosquitoes. Therefore, 3.7% is the proportion of mosquitoes that will bite humans among the total mosquito population.

Using the equation of vectorial capacity, we estimated VC of *Ae. aegypti* for DENV in Reynosa which ranged from 112 to 172 from temperatures ranging between 22 and 30˚C, respectively (Fig 4).

## Discussion

Information on vector-host interactions and associated disease transmission dynamics obtained through the identification of blood meal origin in mosquitoes is important for disease control [4].

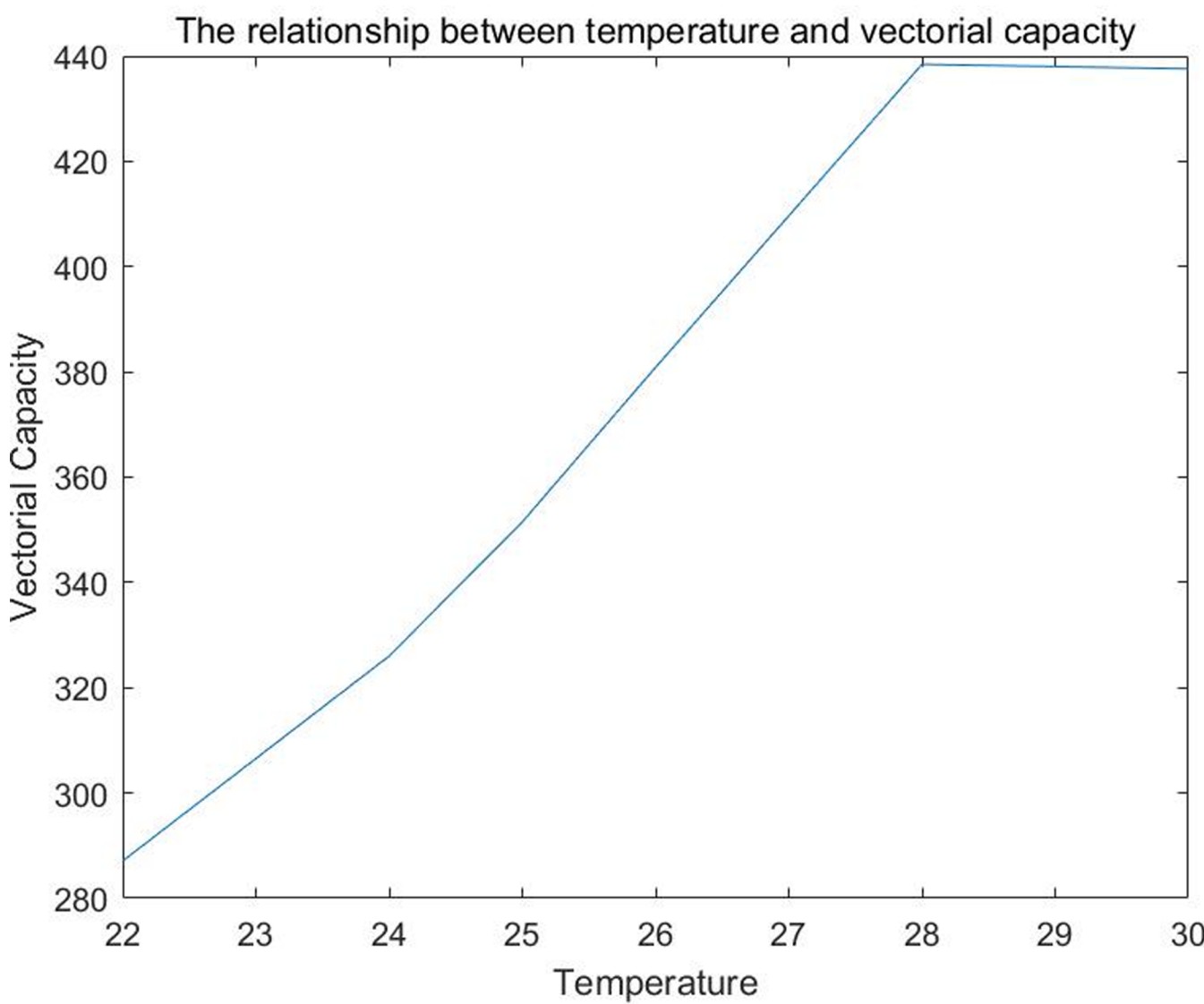

**Fig 4. The relationship between temperature and vectorial capacity of *Aedes aegypti*.** Extrinsic incubation period, vector competence, survival rate, and vectorial capacity were calculated using models as reported elsewhere [31–34]. It should be clarify that the vectorial capacity model used is for the mosquito population in general, instead of an overall model of vectorial capacity for *Aedes aegypti*.

The acquisition of a blood meal is an essential component of the vectorial potential of any mosquito vector. Also important is whether the female mosquito preferentially feeds on animals or humans, particularly in estimating the basic reproduction number associated with the transmission potential of the system under study. This, in turn, is closely linked to vector control strategies. In this context, the occurrence of epidemics involving viruses transmitted by *Ae. aegypti* have increased in the Americas. In the present study, we observed an overall relatively low density for *Ae. aegypti* in residential neighbourhoods of Reynosa, relative to *Cx. quinquefasciatus*. An earlier study in the Nuevo Amanecer neighbourhood in the eastern tip of Reynosa city, reported five host-seeking *Ae. aegypti* per household [36], a stark contrast to 20 females/household reported in a dengue-endemic tropical country of Thailand [38]. Although it is arguable that a low abundance of *Ae. aegypti* might reduce disease transmission, there is limited knowledge as to how this influences the ability of *Ae. aegypti* to feed on humans in highly urbanized areas of Northern Mexico, and how proportional human feeding might drive virus transmission potential.

*Aedes aegypti* is considered a highly anthropophilic species [39–42]; however, we found that 61.7% (129/209 examined; S3 Table) of individual *Ae. aegypti* with a blood meal result had fed on non-human hosts. A similar study conducted on the opposite side of the US-Mexico border in South Texas recently found a similar result, with 70% of *Ae. aegypti* feeding on non-human hosts [43]. In the current study, the proportional feeding of *Ae. aegypti* on domestic dogs (54.2%) was almost twice that of the second most common host, humans (28.0%). This was also similar to the findings of the study north of the US-Mexico border where dogs were responsible for 50% of the *Ae. aegypti* host identifications [43]. However, the blood meal analysis methodology of the Olson *et al.* [43] study was PCR-Sanger sequencing which is different to the meta-barcoding next-generation sequencing approach used in the current study, making the results difficult to compare.

Equally surprisingly is the host feeding pattern observed in *Cx. quinquefasciatus*, with domestic dog as the dominant host species. This finding was unexpected because prior studies have shown that *Cx. quinquefasciatus* is strongly ornithophilic [8,44]. Estimating the relative density of wild and domestic animals is difficult, and although our study found that chicken was the second most fed on host by proportion, forage ratio analysis revealed dog, Virginia opossum, and turkey as the most over-utilized hosts. The domestic dog was over-utilized by *Cx. quinquefasciatus* even more so than *Ae. aegypti*, with a 20-fold (79.2%) increase over the second most abundant host, chicken (3.7%). Our forage ratio analysis was based on a survey of animals present in these communities. Given that we were attempting to sample a combination of wild and domestic animals, we used a combination of a 'census', by counting all animals observed during point counts as well as a 'survey', in which a subsample of the animals present was estimated. We asked the homeowners for the number of animals in each household. The forage ratios are based on the relative abundance of these vertebrates, which would be difficult to compare wild animals (such as birds) to domestic animals (such as dogs) given the difference in survey methodologies. The interpretation of the high forage ratio on dogs is difficult to interpret for this reason. Prior published studies report that *Culex quinquefasciatus* are ornithophilic [45] so the difference compared to the current study could be due to host availability. The Olson et al. [43] study conducted in South Texas found that only 22% of the *Cx. quinquefasciatus* blood meals came from dogs while 67% were from chicken and 9% were from wild birds. In the South Texas low-income residential study sites, dogs made up 31% of the vertebrates surveyed in the neighborhood and chickens made up 17% [43]. In the current study, dogs accounted for 11.8% of the vertebrates and chickens accounted for 6.7%, although the survey in the current study included wild birds which were not included in the South Texas study. Considering the slightly different survey designs, the availability of dogs versus chickens appear similar in the low-income communities north and south of the US-Mexico border.

Mixed feeding patterns were also observed in common hosts: 57 (39.3%) *Ae. aegypti* out of 145 individuals examined fed on humans (30 individuals contained single feeding with human blood and 27 individuals contained mixed feeding with human blood; Figs 2 and 3). Very limited studies have reported a feeding pattern in *Ae. aegypti* where humans represent less than half of the blood meals (but see [34]). To our knowledge, this is the first study using next-generation sequencing for the blood meal analysis of *Ae. aegypti*, making it difficult to compare these results to prior published studies. Prior studies on *Ae. aegypti* blood meal analysis often utilized precipitin or ELISA assays, however, ELISA has been reported to have less specificity with assays involving human, dog, and cat blood meals [46]. Additionally, PCR and direct sequencing of the amplicon could miss blood meals containing more than one vertebrate species. Therefore, previous studies likely underestimated the frequency of *Ae. aegypti* feeding on multiple vertebrate species. Additionally, multiple feeding on different individuals of the same species likely occurs frequently, but our methodology would not have detected these observations. Although relatively still expensive in resource-limited settings, advances in next generation sequencing techniques and higher taxonomic coverage in reference databases have enabled the exploration of the meta-barcoding approach for a more reliable analysis of mosquito blood meals.

*Anopheles gambiae* was shown to exhibit polymorphism for host preference [47], and histological examination revealed that nearly one-half of all engorged *Ae. aegypti* females in Thailand contained more than one blood meal, while about one-third of engorged females in Puerto Rico contained multiple blood meals [41]. Although the results of the present study cannot be explicitly compared with these earlier studies [41,47] because constraints of methodology applied; nevertheless, nearly one-quarter (38/145) of *Ae. aegypti* blood meals were mixed and accurately identified by next-generation sequencing technology. Females of *Ae. aegypti* have been shown to take blood meals 2–3 times per gonotrophic cycle, seeking and rapidly taking another meal, two to five hours after taking the first one [41,48]. This probably explains the high proportion of mixed dog-human feeding observed in the present study.

A plausible explanation for the low anthropophily observed in *Ae. aegypti* could be the relatively low human density compared to dogs. Only three of the households surveyed for the animal census were without a pet dog, and that was nullified by the number of stray dogs roaming the streets of the Pedro J. Méndez neighbourhood. This explanation is corroborated by the observation that *Cx. quinquefasciatus*, a species ubiquitously classified as ornithophilic [8, 44], also fed proportionally more on dogs than avian hosts. 304 (78.3%) *Cx. quinquefasciatus* out of 388 individuals examined fed on dogs (254 individuals contained single feeding with dog blood and 50 individuals contained mixed feeding with dog blood; Figs 2 and 3). Indeed, the relative availability of vertebrate hosts can profoundly influence the feeding patterns of host seeking mosquitoes [49]. Although, the estimated abundance of humans in the study area (14.4%) was higher that observed for dogs (11.8%; Table 2), it is significantly lower when compared with the relative abundance of all non-human hosts estimated. Host selection pattern in *Cx. quinquefasciatus* has been suggested to follow the domestic environment [8], which is expected to be predominantly non-human in this study.

*Aedes aegypti* evolved as an effective vector as part of forest-dwelling mosquito group with resting preferences on tree tops and breeding preferences in accumulated water in tree holes. Domestication of these species via the replacement of natural larval habitat with artificial container habitats in the urban landscape resulted in the adaptation to the human environment, and selected for highly endophilic and endophagic behaviour [41,50]. However, *Ae. aegypti* has spread throughout much of the tropical and subtropical regions of the world, with very unique urban landscapes and very different degrees of humans as viable hosts. For example, housing quality and characteristics are known to influence the ability of mosquitoes to enter

homes [15]. Additionally, continuous application of insecticides to households using ultra-low volume adulticides, likely further limits the endophilic and endophagic behaviour in certain regions. Although not explicitly investigated in the current study, fewer mosquitoes were collected indoors using the CDC backpack aspirators, as a proxy for indoor collection, than any of the other traps placed outside (S2 Table). Elizondo-Quiroga *et al.*[8] also observed a higher density of mosquitoes when collected outdoors than indoor in Nuevo Leon. Likewise, a two-year surveillance study of *Ae. aegypti* in Texas showed that outdoor collections in low-income neighbourhoods were almost eight times that of indoor collections, with this ratio increasing to as high as 15 times in middle-income communities [15]. A small change in host defensiveness has been predicted to drive a larger change in host choice [49]. Mosquito blood meal sources often occurring in aggregation are sometimes density-dependent, and anthropophagous insects tend to select for less defensive hosts, a host choice selection mechanism reinforced through limited learning [41,49]. Selection for host preference after insecticide application has been reported in *Anopheles gambiae*, with accompanying behavioural change from indoor to outdoor resting, and feeding on cattle and other non-human hosts [47].

The neurological behaviour underlying mosquito feeding preferences is not well understood, and few efforts have been made to understand the genetic basis [41,47]. It is not audacious to speculate that host selection and preference could be regulated by as yet unidentified genes, which have undergone positive selection in response to conditions that threaten survival.

The low incidence of human feeding of *Ae. aegypti* observed in the present study is significant with regards to disease transmission dynamics and control. In 1970, Hess & Hayes [51] proposed that the concept of zooprophylaxis, wherein non-human mammals available as hosts can reduce bites to humans by dengue vectors, thus reducing the arbovirus transmission potential. This occurrence has been demonstrated in Kenya and Uganda where non-human host feeding by *Ae. aegypti* was considered responsible for this species being a poor vector of yellow fever virus [52]. The suggestion to genetically modify vectors in order to eliminate the chemical signals that promote human-feeding preference has not been seriously considered due to the uncertainty of potential inherent risk [41], leaving zooprophylaxis as a safe and viable option. The utilization of non-human hosts and exophilic feeding observed in *Ae. aegypti* populations in Reynosa and in the South Texas study [43] suggests that zooprophylaxis could be a potential control measure to consider. In places where dogs contribute substantially to *Ae. aegypti* feeding, treating dogs with anti-parasitic drugs would not only protect them from ecto- and endo-parasites, but could also reduce vector populations and reduce the risk of human exposure to arboviruses. For example, ivermectin has been reported to be a safe, short-term insecticide useful for malaria control in areas where mosquitoes are exophagic or exophilic [53]. In some cases, mosquito populations were reduced up to 80% after feeding on ivermectin treated individuals [54]. We also support our argument that anti-parasitic drugs can have an effect on vector populations in the context of the same vector/virus system. For instance, Deus *et al.* [55] found that ivermectin induces *Ae. aegypti* adult mortality and decreases the hatch rate of their eggs.

Diversion of vector bites to non-human competent hosts reduces viral amplification and reduces potentially infectious bites to humans by dengue vectors, thus could be used as a control tool to limit arbovirus transmission. Although an earlier study in a similar Reynosa neighborhood isolated CHIKV from patients with dengue-like symptoms [56], and co-circulation of DENV1-3 and West Nile virus [57], all pools of host-seeking *Ae. aegypti* were negative for the presence of arbovirus RNA (DENV 1–4, ZIKV, and CHIKV), and all pools of *Cx. quinquefasciatus* were negative for WNV RNA. The explosive arrival of viruses transmitted by mosquitoes, such as CHIKV and ZIKV, into the western hemisphere has demonstrated their rapid

ability to become endemic, with significant public health consequences. These viruses are of public health significance in Mexico, and are zoonotic with transmission cycles involving wild or domestic animals. Enzootic transmission cycles with wild mammal reservoirs can sustain virus populations in nature and initiate new foci of urban transmission through spill-over events. As *Ae. aegypti* is a vector of DENV, CHIKV, and ZIKV, while *Cx. quinquefasciatus* is known vector of WNV, we can hypothesize that domestic dogs can serve as a sentinel species for predicting virus activity and human disease risk in Northern Mexico, given the observation of high host utilization of dogs. Canine sero-surveillance has been suggested as a useful tool for public health risk assessment [58]. A study in France and Africa demonstrated the usefulness of dogs as a sentinel for studying WNV epidemiology and circulation [11], and similar studies have been conducted to study Chagas diseases in Texas and Northern Mexico [58,59]. Several studies have suggested dogs can be a sentinel for WNV [11]. Thus, although dogs may be unable to achieve the infectious viremia of arboviruses, seroconversion and neutralizing antibody production increases the opportunity of detecting low level arbovirus transmission.

Here, VC of *Ae. aegypti* for DENV ranged from 112 to 172 depending on temperatures (between 22 and 30˚C, respectively). However, if h were 100% (*i.e.* all *Ae. aegypti* biting humans; 97% of *Ae. aegypti* feeding on humans was documented in Thailand and Puerto Rico) [41], VC would increase 2.5 times (ranging from 287 to 437, respectively). Hence, if the proportion of *Ae. aegypti* bloodmeals on humans were even higher than that reported here, the transmission of dengue viruses in this area would likely be much higher. Pooled mosquitoes would not have contained blooded individuals to avoid false positive results when modeling the VC due to undigested blood meal. The vector competence is the product of several factors, and should be varying with time. For example, the factor IM(t)/N_M(t), which represents the prevalence of dengue virus infection should be varying with time. We assumed that it is a constant due to the absence of such data. Similarly, the ratio of the number of new cases to the number of humans is also assumed to be constant. Some parameters that are unavailable were derived from literature, such as human biting rate, a, and the ratio of the number of mosquitoes to the number of humans are also estimated assuming that there are four people in one household. The survival rate varying with time is also from literature. The estimations would be more accurate with more data available.

In conclusion, the population of *Ae. aegypti* in Reynosa, Mexico fed on humans only 39.3% of the time, probably due to the abundance of non-human hosts in the residential neighbourhood and the low human density. The high rate of non-human blood meals for *Ae. aegypti* and dog blood meals for *Cx. quinquefasciatus* suggests that zooprophylaxis could be a control option to consider for DENV, ZIKV, CHIKV, and WNV, respectively. Nonetheless, the high number of blood meals from dogs and cats is in of itself a public health concern, due to the concomitant increase in the likelihood of zoonotic pathogen transmission (such as canine heartworm) and the potential spill-over to human populations. This study identifies the need to re-evaluate our knowledge of host utilization by *Ae. aegypti* and *Cx. quinquefasciatus* by meta-barcoding-like approach with next-generation deep sequencing technology in other regions of Mexico and the tropics in general; Determining VC´s components would be useful determinants to help tailor successful vector control programs.

## Supporting information

**S1 Table. Vertebrate-specific primers used for first round PCR.** The sequences of the M13F and M13R tails are in bold, while the COI-specific sequences are in regular font. The M13 tails served as second round PCR primer binding sites, on which Ion Torrent sequencing adapters and UMIs (IonXpress 1–96) were fused.
(DOCX)

**S2 Table. Collection data of mosquitoes collected in Reynosa in May-June and September-November 2018 for this study.**
(DOCX)

**S3 Table. Number of read counts for each host identification in individual *Aedes aegypti* and *Culex quinquefasciatus* examined through next-generation deep sequencing.**
(DOCX)

**S4 Table. Host population densities documented by survey in the study area of Pedro J. Mendez neighbourhood, Reynosa, Northern Mexico.**
(DOCX)

## Acknowledgments

We are grateful to the residents of the neighbourhood Pedro J. Mendez in Reynosa that granted permission to trap mosquitoes on their properties. We thank Lihua Wei, Janet J. Flores-García, Ricardo Palacios-Santana, Irma G. Cobos-Sosa, and Cristian G. Delgado-Corona for assistance in the field work. We thank Wendy Tang and Estelle Martin for assistance with testing mosquito pools for viruses. We thank the World Reference Center for Emerging Viruses and Arboviruses (WRCEVA) at the University of Texas Medical Branch and the Centers for Disease Control and Prevention for providing the dengue virus positive controls used in this study.

## Author Contributions

**Conceptualization:** Paul D. N. Hebert, Gabriel L. Hamer, Mario A. Rodríguez-Pérez.

**Data curation:** Nadia A. Fernández-Santos, Adeniran A. Adebiyi, Sean W. J. Prosser.

**Formal analysis:** Adeniran A. Adebiyi, Sean W. J. Prosser, Anthony R. Fooks, Gabriel L. Hamer, Ling Xue.

**Funding acquisition:** Paul D. N. Hebert, Gabriel L. Hamer, Mario A. Rodríguez-Pérez.

**Investigation:** José Guillermo Estrada-Franco, Nadia A. Fernández-Santos, Adeniran A. Adebiyi, María de J. López-López, Jesús A. Aguilar-Durán, Luis M. Hernández-Triana, Sean W. J. Prosser, Mario A. Rodríguez-Pérez.

**Methodology:** José Guillermo Estrada-Franco, Nadia A. Fernández-Santos, Adeniran A. Adebiyi, María de J. López-López, Jesús A. Aguilar-Durán, Luis M. Hernández-Triana, Sean W. J. Prosser, Gabriel L. Hamer.

**Project administration:** Nadia A. Fernández-Santos.

**Resources:** Anthony R. Fooks.

**Supervision:** Nadia A. Fernández-Santos.

**Validation:** Sean W. J. Prosser, Ling Xue.

**Visualization:** Sean W. J. Prosser.

**Writing – original draft:** Adeniran A. Adebiyi, Mario A. Rodríguez-Pérez.

**Writing – review & editing:** Adeniran A. Adebiyi, Sean W. J. Prosser, Paul D. N. Hebert, Anthony R. Fooks, Gabriel L. Hamer, Mario A. Rodríguez-Pérez.

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
