## [Decision Letter · Decision Letter 0]

16 Jun 2020

Dear Dr. Rodriguez-Perez,

Thank you very much for submitting your manuscript "Vertebrate-Aedes aegypti and Culex quinquefasciatus (Diptera)-arbovirus transmission networks: Non-human feeding revealed by meta-barcoding and next-generation sequencing" for consideration at PLOS Neglected Tropical Diseases. As with all papers reviewed by the journal, your manuscript was reviewed by members of the editorial board and by several independent reviewers. In light of the reviews (below this email), we would like to invite the resubmission of a significantly-revised version that takes into account the reviewers' comments. 

We cannot make any decision about publication until we have seen the revised manuscript and your response to the reviewers' comments. Your revised manuscript is also likely to be sent to reviewers for further evaluation.

Sincerely,

Pattamaporn Kittayapong, Ph.D.

Associate Editor

Rebecca Rico-Hesse

Deputy Editor

Reviewer's Responses to Questions

**Key Review Criteria Required for Acceptance?**

**Methods**

-Are the objectives of the study clearly articulated with a clear testable hypothesis stated?

-Is the study design appropriate to address the stated objectives?

-Is the population clearly described and appropriate for the hypothesis being tested?

-Is the sample size sufficient to ensure adequate power to address the hypothesis being tested?

-Were correct statistical analysis used to support conclusions?

-Are there concerns about ethical or regulatory requirements being met?

Reviewer #1: The work by Estrada-Franco and collaborators contain a reasonable sample size coupled with a robust metabarcoding-like approach/analysis. This strategy was, to my criteria, effective in answering the manuscript's key question, namely the blood meal source of two main arboviral vectors: Aedes aegypti and Culex quinquefasciatus. 

As for the methods section, I have the following specific questions:

1) Would the authors mind to elaborate on the existence or necessity of an ethical committee's approval for the questionaries applied in the households of Reynosa? 

2) Line: 239: "...DENV was tested..."

Did DENV detection also include a positive control, similarly to what was used for ZIKV and CHIVK?

Reviewer #2: All above were adequately addressed. Not sure if IRB needed for household interviews i fdone.

**Results**

-Does the analysis presented match the analysis plan?

-Are the results clearly and completely presented?

-Are the figures (Tables, Images) of sufficient quality for clarity?

Reviewer #1: The metabarcode-like approach and the analyses associated with such methodology were clearly transcribed into a fairly accessible way in the results section, in which I have only a few suggestions/concerns. With that being said, it is clear that for both vectors here studied, humans were not the main source of blood. A rather interesting result, particularly when looking at mixed blood meals. 

As for the results section, I have the following specific questions:

Line 367: "A total of 2,941 females Ae. aegypti were collected during the study."

1) Were any of these samples tested for a mosquito endogenous control gene, as to check for sample integrity?

2) After reading the methods section, I believe that all mosquitoes tested here did not contain a blood meal, correct? Can the authors add a few sentences on the rationale behind this selection approach in the discussion? This is a good opportunity to educate the readers on the importance of experimental design in the context of false positive results in vector competence studies due to undigested blood meal. 

Lines 379-380: "According to the findings of Rodríguez-Pérez et al.34, six people were IgM-positive and seroconverted from a total of 77 examined people, therefore the Incidence/SH =6/(77-6)."

3) Is it correct to assume that this study was conducted at the same period in which the referenced work was also conducted? If so, please state. 

Lines 392-394: "Using the equation of vectorial capacity, we estimated VC of Ae. aegypti for DENV in Reynosa which ranged from 112 to 172 from temperatures varing between 22 and 30 ºC, respectively."

4) The word "varing" is misspelled. If chosen to keep the mathematical component of the paper, would the authors please include a plot for the vectorial capacity as a function of temperature given the parameters utilized in this study?

5) As for figures 2 and 3: I find this representation a bit hard to read given the color scheme selected and the amount of groups present. With that being said, I wonder if the authors would be willing to try a circular packing approach to display their data. It would allow to form big categories (circles) that are predominant, such as "dogs" containing inner circles of variable sizes (representing variable percentages) in which the mixed feeding behavior would be represented. The R code for such representation can be found at: https://www.r-graph-gallery.com/circle-packing.html under the "several levels" category.

Reviewer #2: Figure 3 and perhaps 2 could be much improved (the blood meal analyses) by visualizing them as bipartite network mapping with visualization of species instead of that especially confusing pie chart in Figure 3. This should be improved, in my opinion.

**Conclusions**

-Are the conclusions supported by the data presented?

-Are the limitations of analysis clearly described?

-Do the authors discuss how these data can be helpful to advance our understanding of the topic under study?

-Is public health relevance addressed?

Reviewer #1: The data provided by the authors clearly supports the conclusions raised in the manuscript, as well as the provide guidance towards the benefits of such work in advancing the understanding of the topic under study, and its public health relevance. With that being said, I do still have some specific concerns about the discussion/conclusion part of the manuscript, which are outlined bellow:

Lines 427-431: "Although, chicken was the second most fed on host by proportion, forage ratio analysis revealed dog, Virginia opossum, and turkey as the most over-utilized hosts. The domestic dog was over-utilized by Cx. quinquefasciatus even more so than Ae. aegypti, with a 20-fold (79.2%) increase over the second most abundant host, chicken (3.7%)."

1) Could the authors please comment on the potential for the FR analyses to be biased, based on the sampling approach utilized? For instance, in general, birds are known to be more active early during sunrise or sunset, hence, an analyses conducted at around noon would potentially underestimate the abundance in a given area. I understand that it is hard to fully capture the species abundance in a given area, and this is totally acceptable in my point of view, however I few like a few sentences stating this issue should be present here in the discussion. 

Lines 467-468: "A plausible explanation for the low anthropophily observed in Ae. aegypti could be the relatively low human density compared to dogs."

2) Can the authors expand the discussion section to include a paragraph describing the sociodemographic as well as epidemiological status of Reynosa for the arboviruses here assayed? It would give the readers, not familiar with the region, a better understanding of the results here presented. 

I would like to expand on the importance of such information, by giving an example based on a previous work I was involved with. 

In a certain low-income area with high incidence of dengue in the population, we observed that the deployment of our vector control strategy did not have a major impact on case numbers. After a careful sociodemographic evaluation, it became clear to us that the issue was not associated with the vector control strategy by itself, but with how the population dynamics work in that area. 

Given the low-income aspect of the area and the lack of business to provide jobs for its inhabitants, a vast majority of the individuals in that area would spend around >80% of its time, outside the area, working, hence acquiring dengue outside the region in which we deployed our vector control strategy. As such, the result here present makes me wonder if what we observed in our study would also be a potential explanation for the high feeding rate in dogs and other hosts rather than humans. For instance one is to expect dogs to spend all day in a certain area whereas the same would most likely not be the case for the pet owners/humans, hence the chances of a mosquito encountering a host other than humans to blood feed would be higher. 

Lines 522-525: "For example, ivermectin has been reported to be a safe, short-term insecticide useful for malaria control in areas where mosquitoes are exophagic or exophilic50. In some cases, mosquito populations were reduced up to 80% after feeding on ivermectin treated individuals51."

3) I would avoid mixing studies on parasites with the idea of arboviruses control. If the authors want to support their argument, then please use references in the context of the same vector/virus system. For instance DOI:10.1603/me11164 is a work using the same drug in Ae. aegypti, rather than Anopheles mosquitoes. 

Lines 526-527: "Diversion of vector bites to incompetent arbovirus host maybe responsible for the low

527 transmission rates observed in the study area."

4) I find this sentence confusing and rather difficult to understand. Would the authors mind to rephrase it?

Lines 541-543: "Although dogs may be unable to achieve the infectious viremia of ZIKV, seroconversion and neutralizing antibody production increases the opportunity of detecting low level arbovirus transmission."

5) I would like to ask the authors to please briefly expand on such argument. For instance, before proposing such alternative approach, is there any data in the literature to predict such strategy as a reliable source for Flavivirus surveillance, in the context of antibody specificity and cross-reactivity?

Lines 545-549: "respectively). However, if h were 100% (i.e. all Ae. aegypti biting humans; 97% of Ae. aegypti feeding on humans was documented in Thai and Puerto Rico)39, VC would increase 2.5 times (ranging from 287 to 437, respectively). Hence, if the proportion of Ae. aegypti bloodmeals on humans were even higher than that reported here, the transmission of dengue viruses in this area would likely be much higher."

6) I would recommend removing this hypothetical scenario based on feeding rates observed in other regions and just focus on the results collected by the authors. With that being said, I am not particularly enthusiastic about mathematical models based on data collected elsewhere and I am not convinced that the model here presented adds any relevant information to the work, but rather interfere with the reader's focus on the novelty aspect of this work, which is the bar coding/NGS approach to identify host feeding behavior. My recommendations would for the authors would be to:

A) Completely remove the vectorial capacity component of this manuscript, as it heavily relies on data from other regions/mosquito population genetics which might under or overestimate the results here obtained. 

B) Leave the mathematical model, but include a paragraph expanding on the limitations of the model here used in the context of data availability/collection.

Reviewer #2: The limitations of the analyses could be improved.

**Editorial and Data Presentation Modifications?**

Reviewer #1: All my main scientific concerns, which are the main point of the paper, were already raised in the other sections of the work. I would just suggest a few minor corrections: 

Line 83: "found globally in world..."

Would the authors mind rephrasing it?

Line 94-95: "Various techniques that have been employed for the analysis of mosquito blood meals including ELISA or precipitin tests6, 7, and PCR-based approaches8."

Would the authors mind rephrasing it?

Line 121: "Mosquito"

Please correct it to Mosquitoes

Line 128: "Ovitraps (AGO) traps (Springstar) that was checked weekly15"

Replace was by were

Line 135: "Mosquito identifications was based"

Would the authors mind rephrasing it?

Line 157: "0.5 mMEDTA pH"

Please include space between mM and EDTA

Reviewer #2: See above for figure 3

Minor editorial comment: Line 289 'sampling' should not be italicized

**Summary and General Comments**

Reviewer #1: The work by Estrada-Franco introduce the use of a metabarcoding-like approach to identify the blood meal source of two key arboviral species: Aedes aegypti and Culex quinquefasciatus in the region of Reynosa - Mexico. By using such refined technique, the authors were successfully able to characterize the feeding behavior of these two species, as well as obtain series of parameters derived from these results, namely the vector forage ratio and the human blood index for the region studied. For instance, it is interesting to notice that dogs but not humans were the most over-utilized host by

both Ae. aegypti and Cx. quinquefasciatus. 

I believe that the use of NGS technology and the blood feeding behavior results obtained through such technique for two key arboviral species is already sufficient to qualify this work for publication. The addition of a mathematical model for vectorial capacity, mainly based on parameters estimated by others using different mosquito populations from other regions, does not add any value to the manuscript, but rather remove some of the focus on the results obtained. Given the lack of a proper expansion on the implications of such model in all sections of the paper, particularly in the discussion section, I would recommend for the complete removal of it.

Reviewer #2: This was an interesting survey of blood meal analyses of mosquito vectors in a neighborhood in Northern Mexico and methods appear adequate. However, there are a few questions/comments that I think should be addressed, and it relates primarily to the study site description and field methodology: The neighborhood is poorly described. Is this a rural or urban area? How many people live in the neighborhood sampled? What is the typical household constructed of? What is the general housing and yard infrastructure in the sample area. How large is this neighborhood (how many households in total) and how were these 16 households chosen? How representative was the household sampling of the neighborhood of study? How did you determine they were representative of the range of habitats for the mosquito vectors. What is the relative spatial distribution of these households and does this impact patterns of blood meal feeding patterns and infection even? How permeable are households to mosquitoes? Do inhabitants spend a lot of time outside? 

Please provide a description of household and yards (type, etc) 

I might have missed this but were bugs pooled at the household level or in general at community level when tested for viruses?

Since you saw a relatively low number of mosquito feeding on humans (esp. for quinquefasciatus), did you do any behavioral studies evaluating how often people are resting or active outside around the households of study, or do people spend most of their time indoors in these neighborhoods?

Figure 3 and perhaps 2 should be improved visually (some type of bipartite network0

PLOS authors have the option to publish the peer review history of their article (what does this mean?). If published, this will include your full peer review and any attached files.

Reviewer #1: Yes: Heverton Leandro Carneiro Dutra

Reviewer #2: No
---

## [Decision Letter · Decision Letter 1]

10 Sep 2020

Dear Dr. Rodriguez-Perez,

Thank you very much for submitting your manuscript "Vertebrate-Aedes aegypti and Culex quinquefasciatus (Diptera)-arbovirus transmission networks: Non-human feeding revealed by meta-barcoding and next-generation sequencing" for consideration at PLOS Neglected Tropical Diseases. As with all papers reviewed by the journal, your manuscript was reviewed by members of the editorial board and by several independent reviewers. In light of the reviews (below this email), we would like to invite the resubmission of a significantly-revised version that takes into account the reviewers' comments. 

We cannot make any decision about publication until we have seen the revised manuscript and your response to the reviewers' comments. Your revised manuscript is also likely to be sent to reviewers for further evaluation.

Sincerely,

Pattamaporn Kittayapong, Ph.D.

Associate Editor

Rebecca Rico-Hesse, Ph.D.

Deputy Editor

Reviewer's Responses to Questions

**Key Review Criteria Required for Acceptance?**

**Methods**

-Are the objectives of the study clearly articulated with a clear testable hypothesis stated?

-Is the study design appropriate to address the stated objectives?

-Is the population clearly described and appropriate for the hypothesis being tested?

-Is the sample size sufficient to ensure adequate power to address the hypothesis being tested?

-Were correct statistical analysis used to support conclusions?

-Are there concerns about ethical or regulatory requirements being met?

Reviewer #1: (No Response)

Reviewer #2: -Are the objectives of the study clearly articulated with a clear testable hypothesis stated? For the most part

-Is the study design appropriate to address the stated objectives? yes 

-Is the population clearly described and appropriate for the hypothesis being tested? Yes

-Is the sample size sufficient to ensure adequate power to address the hypothesis being tested? Not sure did they doa power analysiss

-Were correct statistical analysis used to support conclusions? Some additional analyses could be done

-Are there concerns about ethical or regulatory requirements being met? No

**Results**

-Does the analysis presented match the analysis plan?

-Are the results clearly and completely presented?

-Are the figures (Tables, Images) of sufficient quality for clarity?

Reviewer #1: The legend for figure 4 needs more information. Include a reference for the papers in which the models were based, as they are currently buried in the methods section. Specify that the model is for the mosquito population in general, instead of an overall model of vectorial capacity for Aedes aegypti.

Reviewer #2: -Does the analysis presented match the analysis plan?- More or less

-Are the results clearly and completely presented? Not completely 

-Are the figures (Tables, Images) of sufficient quality for clarity? Not completely

Table 2 and 3- Please rank species by forage ratio or % of host fed level in order from highest to lowest, so it is easier to interpret the data. 

Also, Figures 2 and 3 would be easier to read using a bipartite network where blood meal-mosquito associations can be evaluated and visualized in a more elegant way that is clear for the reader to interpret.

**Conclusions**

-Are the conclusions supported by the data presented?

-Are the limitations of analysis clearly described?

-Do the authors discuss how these data can be helpful to advance our understanding of the topic under study?

-Is public health relevance addressed?

Reviewer #1: (No Response)

Reviewer #2: -Are the conclusions supported by the data presented? Nearly completely

-Are the limitations of analysis clearly described? result limitations for the most part adequately addressed 

-Do the authors discuss how these data can be helpful to advance our understanding of the topic under study? Yes

-Is public health relevance addressed? Yes

**Editorial and Data Presentation Modifications?**

Reviewer #1: Please, revise the manuscript one more time, as there are many typos and misspelled words.

Reviewer #2: able 2 and 3- Please rank species by forage ratio or % of host fed level in order from highest to lowest, so it is easier to interpret the data. 

Also, Figures 2 and 3 would be easier to read using a bipartite network where blood meal-mosquito associations can be evaluated and visualized in a more elegant way that is clear for the reader to interpret.

**Summary and General Comments**

Reviewer #1: I am ok with the current version of the manuscript, as well as the answers provided by the authors to all of my concerns. I still do not appreciate the mathematical model component and the way it was performed, however I respect the author's decision to keep it, particularly because I do not see it as absolutely wrong, but far too "vague". I still believe that the NGS approach would be sufficient to qualify the manuscript for publication.

Reviewer #2: This paper provides useful descriptive information about blood meal species composition for Aedes and Culex in the study area. Most of the previous reviewers’ comments appear to be addressed. The results of dogs as important blood meals and the discussion on zooprophylaxis are interesting.

PLOS authors have the option to publish the peer review history of their article (what does this mean?). If published, this will include your full peer review and any attached files.

Reviewer #1: Yes: Heverton Leandro Carneiro Dutra

Reviewer #2: No
---

## [Editor Report · Decision Letter 2]

9 Oct 2020

Dear Dr. Rodriguez-Perez,

We are pleased to inform you that your manuscript 'Vertebrate-Aedes aegypti and Culex quinquefasciatus (Diptera)-arbovirus transmission networks: Non-human feeding revealed by meta-barcoding and next-generation sequencing' has been provisionally accepted for publication in PLOS Neglected Tropical Diseases.

Best regards,

Pattamaporn Kittayapong, Ph.D.

Associate Editor

Rebecca Rico-Hesse

Deputy Editor

---

## [Editor Report · Acceptance letter]

16 Dec 2020

Dear Dr. Rodriguez-Perez,

We are delighted to inform you that your manuscript, "Vertebrate-*Aedes aegypti* and *Culex quinquefasciatus* (Diptera)-arbovirus transmission networks: Non-human feeding revealed by meta-barcoding and next-generation sequencing," has been formally accepted for publication in PLOS Neglected Tropical Diseases.

Best regards,

Shaden Kamhawi

co-Editor-in-Chief

Paul Brindley

co-Editor-in-Chief
